# Evolution of a System to Monitor Infant Neuromotor Development in the Home: Lessons from COVID-19

**DOI:** 10.3390/healthcare11060784

**Published:** 2023-03-07

**Authors:** Manon Maitland Schladen, Hsin-Hung Kuo, Tan Tran, Achuna Ofonedu, Hanh Hoang, Robert Jett, Megan Gu, Kimberly Liu, Kai’lyn Mohammed, Yas’lyn Mohammed, Peter S. Lum, Yiannis Koumpouros

**Affiliations:** 1Department of Biomedical Engineering, The Catholic University of America, Washington, DC 20064, USA; 2Department of Electrical Engineering and Computer Science, The Catholic University of America, Washington, DC 20064, USA; 3Department of Mechanical Engineering, The Catholic University of America, Washington, DC 20064, USA; 4Department of Public and Community Health, University of West Attica, 12243 Aigaleo, Greece

**Keywords:** COVID-19, neuromotor delay, infants, home-based technologies, family-centered technology

## Abstract

In the nine months leading up to COVID-19, our biomedical engineering research group was in the very early stages of development and in-home testing of HUGS, the Hand Use and Grasp Sensor (HUGS) system. HUGS was conceived as a tool to allay parents’ anxiety by empowering them to monitor their infants’ neuromotor development at home. System focus was on the evolving patterns of hand grasp and general upper extremity movement, over time, in the naturalistic environment of the home, through analysis of data captured from force-sensor-embedded toys and 3D video as the baby played. By the end of March, 2020, as the COVID-19 pandemic accelerated and global lockdown ensued, home visits were no longer possible and HUGS system testing ground to an abrupt halt. In the spring of 2021, still under lockdown, we were able to resume recruitment and in-home testing with HUGS-2, a system whose key requirement was that it be contactless. Participating families managed the set up and use of HUGS-2, supported by a detailed library of video materials and virtual interaction with the HUGS team for training and troubleshooting over Zoom. Like the positive/negative poles of experience reported by new parents under the isolation mandated to combat the pandemic, HUGS research was both impeded and accelerated by having to rely solely on distance interactions to support parents, troubleshoot equipment, and securely transmit data. The objective of this current report is to chronicle the evolution of HUGS. We describe a system whose design and development straddle the pre- and post-pandemic worlds of family-centered health technology design. We identify and classify the clinical approaches to infant screening that predominated in the pre-COVID-19 milieu and describe how these procedural frameworks relate to the family-centered conceptualization of HUGS. We describe how working exclusively through the proxy of parents revealed the family’s priorities and goals for child interaction and surfaced HUGS design shortcomings that were not evident in researcher-managed, in-home testing prior to the pandemic.

## 1. Introduction

Infancy is a time of life filled with great anticipation. New parents may vacillate between excitement over the potential waiting to unfold as their babies grow and fear that development may not progress as it should. This ambivalence may be particularly acute for parents of infants identified at risk for delay in the perinatal period. Risk may stem from the occurrence of a pre-term, or complicated, birth, as well as from use of life-saving technologies such as ECMO (extra corporeal membrane oxygenation) to address respiratory or cardiac failure in the infant [1]. 

Increasingly, parents may have been alerted to potential neurodevelopmental problems by concerning findings on MRI (magnetic resonance imaging). In recent years, safe and effective techniques for addressing the problem of infant movement during image acquisition as well as increased ability to handle movement artifacts during post-processing [2] have led to wider use of MRI for children, where the perinatal history suggests a potential neural insult. Infants identified as at risk for developmental delay via objective imaging criteria are typically referred for early therapeutic intervention to minimize impairment as the child grows through infancy and into childhood. For infants whose perinatal history does not raise red flags, a wait-and-see-how-development-goes approach is adopted, with developmental screening typically paired with well-baby visits to the pediatrician during the first year after birth. 

However, neither an uneventful perinatal history nor the absence of abnormal findings on infant MRI guarantees that an infant will not experience neuromotor delay. For example, half [3] of all children who ultimately receive a diagnosis of cerebral palsy (CP), the most common neurodevelopmental disorder of childhood [4], do not experience an identifiable risk in the time surrounding birth. As a result, diagnosis and subsequent intervention do not occur until between 12 and 24 months of age in higher-income countries and not until 5 years of age in less-resource-rich environments [5]. The effect of delayed intervention is that the window of greatest neuroplasticity, infancy, is missed and the potential for optimal developmental outcomes is diminished. 

There are other cases as well where parents may also experience a sense of risk despite an apparently typical pregnancy and birth. Parents of infants who have a sibling with a diagnosed developmental impairment, such as autism spectrum disorder (ASD), fall into this category based on suggested familial risk [6]. As is the current clinical thinking relative to CP, detection of signs of ASD as early as possible [7,8] facilitates the introduction of interventions most likely to be effective in maximizing positive outcomes. 

The SARS-CoV-2 pandemic added yet another layer of uncertainty to the experience of families with infants identified as at risk [9]. For some new parents who had no reason to believe that their newborns were at developmental risk, COVID-19 was reported to further exacerbate the anxiety that typifies the perinatal period and add new fears related to both possible viral transmission and enforced social isolation under lockdown [10]. Conversely, other new parents experienced this withdrawal from life as usual, precipitated by distancing measures, as peaceful, providing an opportunity to consolidate family relationships and realign priorities [11].

Service delivery models shifted to an all-but-universal embrace of telehealth in response to COVID-19 infection control policies [12]. The move to telehealth demanded fundamental changes be made to both the philosophy and practice of pediatric rehabilitation; however, practitioners had no choice but to engage with this challenge [13,14,15,16]. The in-person environment of the clinic is regulated by practitioners, whereas the telehealth environment of the home is under the pediatric patient family’s control. This shift in the service delivery environment provided an unexpected practicum for authentic, relationship-based, family-centered care [17]. 

In the therapies, the accepted practice model is relationship-driven family-centered care, RDFCC [18]: family, versus patient, since patients, particularly pediatric ones, live their lives in continual interplay with the intimate, social units we call family [19]. When the pediatric patient is an infant, the identity of a patient with family is even more pronounced. Among the tenets of the RDFCC model is the importance of working through the parents/family to integrate therapy into a child’s daily routine [20]. The relationship the therapist develops with the family is central, with the therapist functioning as a peer, focused on the priorities and preferences of the family to help them map out a home program for the child personalized to the family’s unique needs. The rationale for this approach is that the program that aligns well with family life is the one that is most likely to be successful. Therapists found that even though they accepted the premises of RDFCC prior to COVID-19, having to work with families from a distance helped them hone their skills and discover better ways of helping families integrate therapy into their child’s daily life [17]. 

Family-centered care has its parallel in the development of health technology for use in the home. A systematic review [21] uncovered 41 distinct frameworks proposed for “patient-/service-user (for example: parents, caregivers, and families)-centered engagement”, PSUE. Generalizing across frameworks, Shippee and colleagues identified common PSUE components of (1) patient initiation, (2) reciprocal patient/researcher relationships, (3) co-learning, and (4) re-assessment and feedback. These components align with the principles of relationality, parity, coaching, and empowerment found in the RDFCC model.

This paper explores the experience of our rehabilitation engineering center (the Rehabilitation Engineering Research Center RERC on Patient-centered, Home-based Technologies to Assess and Treat Motor Impairment in Individuals with Neurologic Injury (See https://engineering.catholic.edu/rerc-dc/, accessed on 26 February 2023)) during the COVID-19 pandemic as we worked to design and develop home monitoring technology for families of infants at risk for neuromotor delay. We call our technology HUGS: the Hand Use and Grasp Sensor system. The purpose of our development was two-fold. First, we wanted to leverage machine learning and applied kinematics to better understand what features of infant hand use in the data-rich, day-to-day, naturalistic environment of the home might signal cause for concern or be reassuring. Second, we wanted to understand what features of a technology would lead to its acceptance by and usefulness to families of infants. If the technology is not used, the information it might theoretically provide is of no benefit. Acceptance, rejection, or ultimate abandonment of technology are all functions of the end user’s perception of both usefulness and ease of use [22,23]. Echoing the theme of the importance of family priorities in the RDFCC model, the usefulness of a technology is tightly linked to its ability to promote outcomes that families themselves find important.

### Technology Concept and High-Level Description

HUGS was conceived as a family-centered, home-based approach to objectively and quantitatively assess the development of hand grasp in infants at risk for fine motor delay, and to predict the future diagnosis of delay. Undiagnosed motor delay can prevent the timely use of early interventions and lead to motor disabilities throughout life. The initial HUGS protocol called for conducting biweekly sessions with parents and infants 12–36 months of age to gather data on spontaneous hand use. Data consisted of video of infants’ movements and grasp force, measured through instrumented toys that were part of the HUGS suite. 

As HUGS toys measured the forces infants generated during play, they simultaneously provided infants biofeedback proportional to the magnitude of their grasp force through three sensory channels: touch (vibration), vision (lights), and hearing (sounds). We were interested to learn whether infants adapted their hand grasp based on a preference for specific sensory stimuli. This information might signal potential avenues for early interventions to encourage object exploration and functional hand use. Researchers have demonstrated that developmental problems in infants can be detected from 3D kinematic analysis of their general movements (GMs) in the lab using complex, multicamera systems [24,25]. Accordingly, in addition to measuring infant grasp force, the HUGS system also explores the degree to which new, highly compact and portable 3D cameras might lead to in-home versus in-lab infant motion capture and subsequent kinematic analysis.

## 2. Methods and Materials

### Organization of This Paper

Three iterations of HUGS development have taken place over the course of the rise and progression of the COVID-19 pandemic. In the sections that follow, we describe each of our three successive HUGS builds: HUGS-1 (July 2019–March 2020: Pre-lockdown), HUGS-2 (April 2021–November 2022: Mid-lockdown through early normalization), and HUGS-3 (January 2023–forward: the “new normal”, (Wikipedia attributes the first use of this now common phrase to Henry A. Wise Wood writing in the National Electric Light Bulletin in 1918 [26]) post-COVID-19). We describe our initial thinking about the design of HUGS, which was realized in the system we call HUGS-1 and piloted with families of nominally typically developing (TD) infants recruited from the community. Though it was put into the home prior to the ensuing COVID-19 state of emergency, we describe HUGS-1 in detail, as subsequent HUGS builds are grounded in this initial conceptualization. We briefly recount the results of our HUGS-1 testing experience with TD infants in the home and the design modifications that emerged. These modifications are ones that would have been implemented in the follow-on build of HUGS, irrespective of the fundamental shift in protocol that was needed if we were to be able to resume in-home testing during the COVID-19 lockdown. 

Subsequently, we detail the re-working of the HUGS protocol that allowed us to place HUGS-2 in the home during lockdown, under total control of the parent, while we provided tele-coaching and troubleshooting. We describe the testing experience and data outcomes of the five families who tested HUGS when COVID-19 restrictions on community interactions were at their height. We relate how the intensity of parental involvement in HUGS-2 altered the trajectory of HUGS research and shaped the requirements to be implemented in HUGS-3, the HUGS-2 successor prototype. We use the RDFCC model’s framework to reflect on the opportunity for accelerated learning that the circumstances of the COVID-19 lockdown afforded. Finally, we discuss the implications of our experience for family-centered systems design.

## 3. HUGS-1 (July 2019–March 2020, Pre-Lockdown, COVID-19)

### 3.1. The Start of HUGS: Inaugural System Design as HUGS-1

The HUGS study can be characterized as an effort embodying both neuromotor development research and infant home system development goals. From the neuromotor research objective stance, HUGS was conceived to provide data on infant grasp force and upper extremity movement and patterns of engagement, such that hand use developmental trajectories might be developed to more promptly identify infants at risk for delay and provide therapeutic intervention. It was also conceived as the initial exploration of how the data needed to identify infants at risk could be effectively and easily gathered by parents in the course of day-to-day interaction with their infants in the home. 

HUGS-1 consisted of two instrumented toys (the bar toy (Table 1, pane a) and the candy toy (Table 1, pane b)), which could be hung from an A-frame (Figure 1), placed so that the infant, seated in a standardized (and easily cleanable) infant seat (Figure 2), could easily reach the toys. The toys were embedded with force-sensing resistors (FSRs) and wired through the A-frame to an array of Arduino^®^ Uno microcontrollers housed in an electronics box. A dial mounted to the electronics box (Figure 3) controlled functions of the Arduino array, which routed signals to and from the A-frame/toy complex. One of the initial HUGS research hypotheses was that the type of sensory feedback babies received when they handled a sensorized toy would influence the strength and duration of their grasp. As a result, the protocol called for grasp testing in three modes: visual (flashing colored lights), auditory (music), and haptic (vibration). Parents/researchers turned the dial to select (simple randomization) the stimulus to attract the infant to explore the offered toy. 

Two cameras were additionally part of the HUGS-1 configuration. An Intel^®^ RealSense™ 3D camera, connected to and controlled through software on an adjacent laptop (Figure 4), captured infants’ spontaneous, upper extremity movements for later kinematic analysis. An ancillary RGB camera (GoPro), collocated with the RealSense on a bar suspended between two adjustable tripods, provided a second video record of the HUGS-1 interaction. The GoPro feed was principally used to verify infant versus parent or researcher grasp force registered by the FSR as well as to explore whether grasp force varied by hand position. Table 1, pane c shows five grasp patterns typical of infants interacting with the bar toy.

Figure 5 provides a high-level view of the HUGS-1 components and how they worked together. See Appendix A for a copy of the user manual that was shared with parents.

### 3.2. Putting HUGS-1 into the Home

Ethical approval for the HUGS study (#19-0012, Assessing Infant Motor Development in the Home: AIM Home) was obtained from the Institutional Review Board (IRB) of the Catholic University of America, Washington, D.C., as well as from the IRB (IRBear # Pro00011884, Identifying Patient- and Caregiver-Centered Outcomes to Guide the Design and Implementation of Technology for Infant Motor Development Assessment in the Home) of our adjacent clinical partner, Children’s National Hospital. Written informed consent was obtained from one of each participating infant’s parents prior to setting up the HUGS-1 system in the home, subsequent testing, and data collection. Families with infants 3 to 36 months of age, presumably typically developing (TD), were recruited from the community to pilot HUGS-1. 

Several changes to the HUGS-1 home testing protocol as well as to the prototype’s form factor were implemented over the course of piloting. As our appreciation of computer vision and identification of infant upper extremity points of interest (PoI) from video became more nuanced, we shifted from a marker-based approach to PoI identification to a markerless one. There was no longer any need for special garments or simulated green screen backgrounds. Removal of both requirements simplified the process of getting HUGS-1 set up and ready to capture data. The need for simplification was apparent during interactions with the first several families enrolled. Visual inspection of Figure 5 demonstrates that HUGS-1 had a lot of moving parts. Within the first several sessions, one family withdrew from the study and several confided that they were thinking of withdrawing due to burden and complexity. Parents’ desire to withdraw curtailed the original plan that data would be collected without research team presence. When we later return to this idea as a strategy to continue testing despite COVID-19 restrictions, we did so with some experience of the hurdles that would be involved.

The need for researcher presence also added pressure to the timing of data capture. Sessions were scheduled at family convenience, in anticipation of infants being wide awake and affable; however, infants did not always behave as parents expected once the research team arrived. Given social norms of empathy and responsibility typical of adults, parents would encourage moving forward with the session if the infant was at all cooperative, i.e., not asleep or crying. A common parental remark on reviewing a session was that the baby’s grasping seemed happenchance as opposed to motivated and goal-directed. 

The HUGS-1 toys, as pictured in Table 1, are the descendants of earlier variants. See Appendix A for a description of this early evolution. HUGS-1 testing began with a bar toy that was suspended via a square where the baby grasped an FSR-embedded horizontal member to activate lights on the vertical members of the square when visual mode was selected (see Figure 6). The toy design changed to integrate the lights into the bar itself (see Table 1, pane a) on the theory that it would be easier for the baby to associate light with touch if the light occurred where the touch did. Finally, a single-arm tripod (see Appendix A) was exchanged for a more stable two-tripod support with a bar suspended between them where the cameras could be mounted.

### 3.3. Outcomes

When we were forced to abruptly shut down HUGS-1 testing due to the state of emergency precipitated by COVID-19 in March of 2020, we had been able to pilot the system with 12 different infant subjects (7 female). For these infants, we recorded 29 sessions capturing infants’ spontaneous, upper extremity movements and logged a total of 2832 grasp events across 62 HUGS interactions.

#### 3.3.1. Spontaneous Movements

All infants enrolled in the pilot were nominally TD, recruited from the community, with no developmental problems of which parents were aware or communicated. However, study clinicians, specialists in pediatric neurology and physiatry, noted that the presumption of TD was difficult to confirm on video due to infants slipping into positions in the standardized infant seat that made it harder for them to choose to move one arm or the other unimpeded [27]. This observation mirrored comments from HUGS-1 parents that their babies’ movements, particularly before babies’ trunk control, was well developed, seemed awkward. 

#### 3.3.2. Grasp Events

Due to our small sample size, missing data, and data captured at unequal time points, we chose linear mixed modeling (LMM), a technique robust to such inconsistencies [28], to explore infant grasp. We were able to detect a positive change in grasp force and duration, as well as in the number of bimanual grasps, as babies grew older [29]. This finding provided basic validation of the concept that grasp magnitudes and infant age are linearly related. Stimulus mode had no effect on forces gathered from either toy. With respect to the bar toy, neither orientation of the infant’s hand nor its location on the bar had any measurable effect on grasp parameters. 

Infants did not, or were not able to, grasp the candy toy (toy 2) as forcefully as they did the bar toy (toy 1) (see Figure 7). Note that the “bump” in grasp force occurs at six months, as measured by the bar toy, but not until eight months measured by the candy toy. Infants’ weaker performance on the candy versus bar toy was mirrored in recorded duration and frequency of grasp as well. 

## 4. HUGS-2 (April 2021–November 2022: Mid-Lockdown through Early Normalization)

The immediate post-lockdown period was one of extreme uncertainty. IRB imposed a moratorium of unknown duration on all research in progress. Our sponsors graciously extended support. We analyzed the data we had collected on HUGS-1 (previous section) and, by fall 2020, had begun the redesign effort that would have followed in the typical course of technology development. Section 4.1 describes the changes to HUGS-1 brought to light by our first in-home pilot cut short by the pandemic. Section 4.2 describes the further changes made necessary to allow for resumption of in-home testing in the face of COVID-19 distancing requirements.

### 4.1. Design Changes Post-HUGS-1 Testing, Irrespective of COVID-19

The user experience of HUGS-1, along with the infant development findings it mediated, needed to find expression in the next iteration of research and development. Criteria that might be required address conducting research in infants’ homes under lockdown would be considered in tandem with problems that emerged during home testing under the pre-COVID-19 scenario. Four principal modifications, as follows, were identified for HUGS-2.

#### 4.1.1. The Candy Toy

The candy toy was designed to capture variations in palmar grasp. However, infants’ performance on the candy toy versus the bar toy was consistently lower in terms of frequency, force, and duration. The candy toy presented a smaller target for grasp than did the bar toy and less area to explore if the infant actually acquired it. This ontological problem, coupled with the usability problem of greater complexity (i.e., the need to swap in a second toy during a testing session with a baby growing restless), led to the decision to retire the candy toy.

#### 4.1.2. The Standardized Infant Seat

The infant seat we selected to standardize testing across babies was not of a type that has been widely used by families in recent years. All families enrolled in HUGS used a soft, sling-type infant seat in the early months and a variety of seats as the infant gained trunk control. We modified the protocol to allow families to use their infant’s usual seat going forward when testing HUGS-2.

#### 4.1.3. The Ancillary Camera

The main purpose for incorporating a second camera (GoPro) into the HUGS-1 protocol was to capture audio; the RealSense™ has no audio capture capability. We believed audio would provide greater clarity on any difficulties experienced during sessions conducted by parents. We also conceived of using the second feed to verify hand position during grasp force testing. Since we discontinued family solo sessions early in HUGS-1 testing, and the second camera added considerable session management overhead, we decided to drop the second camera and to rely on the 2D feed from the 3D RealSense™ to verify hand placement. We further modified the protocol to ask parents to reposition the baby so the camera captured video from behind during grasp sessions. Most grasps around the bar captured from the RealSense™ were with hands pronated, and we experienced difficulty clearly identifying anatomical landmarks (e.g., the different fingers) when hands were imaged from the front.

#### 4.1.4. Sensory Feedback Modes

One of the initial hypotheses of HUGS was that the type of sensory feedback babies received when they handled a toy—visual, auditory, or haptic—would influence the strength and duration of their grasp. This hypothesis was not supported by our analysis of data from the HUGS-1 pilot study [29]. However, the finding that infants did not modulate their grasp force or time to demonstrate a preference for auditory, visual, or tactile sensory feedback does not mean that sensory feedback in general has no effect on a child’s interest in manipulating a toy. Notably, the Hand Assessment for Infants (HAI), a clinical tool whose validation data were published after the initial development of the HUGS protocol [30], is focused intently on the selection of a range of testing toys to assure that infants receive variable stimulation to ensure that they are engaged and intentional about hand use while being assessed. Therefore, we retained the feedback modes and sequential testing of grasp in response to auditory, visual, and tactile modes in HUGS-2. 

### 4.2. Design Changes to Enable HUGS-2 Testing under Lockdown

By fall of 2020, we began to plan for changes to the HUGS protocol and technology platform that could allow us to resume our work with the next iteration system: HUGS-2. The key challenge was to conduct contactless testing of a system that previous participants had told us they did not want to test without researcher presence. Our strategy to resume testing was to follow in the path of our pediatric clinical colleagues. Whereas they developed procedures to continue to provide clinical care at a distance, leveraging telehealth technology, we developed parallel procedures to leverage distance technologies to support HUGS testing in the home. Table 2 maps the evolution of HUGS to adapt to the health safety demands of COVID-19. The table summarizes the design differences between HUGS-1 and HUGS-2 and identifies whether they were responsive to the experience of the HUGS-1 pilot or anticipated the challenge of resuming in-home testing while observing COVID-19 precautions.

The original plan for HUGS-2 was to test the system with at-risk versus TD infants. We hoped that the prospect of conducting testing autonomously with no in-person contact with the research team might seem more reasonable and palatable to parents of at-risk versus TD infants given the greater presumed fragility and vulnerability of at-risk infants, irrespective of the specific threats of COVID-19. The disadvantage of testing with at-risk infants was that there was an intrinsically smaller pool of potential participants to draw. Infants with serious risk factors for neuromotor delay would, however, continue to be admitted to the hospital where they could be identified and their interest in participating in HUGS-2 testing queried by our clinical partners. 

#### 4.2.1. Streamlining the Conduct of HUGS Sessions

Foremost among the system changes needed to support autonomous use was to make HUGS easier for parents to set up and operate. Compared to HUGS-1 (see Figure 5), HUGS-2 incorporated greater technical complexity but presented the user with fewer, disjointed tasks. Figure 8 sets out the components of the HUGS-2 system and how they worked together. Greater technological complexity is exemplified by the replacement of written instructions to the user for navigating the various tasks involved in collecting HUGS data with a custom GUI and automated data upload process using commercially available technology (i.e., Microsoft OneDrive).

From the perspective of parent burden, the setup and connection of HUGS-2 components only differed from what was required for HUGS-1 in the elimination of the need to collect two sets of grasp data on two toys that the retirement of the candy toy brought about. As in HUGS-1, the toy bar/frame connected to the electronics box, which, in turn, was connected to the USB-2 port of the study laptop (see Figure 9). Additionally, since HUGS-2 needed to be packaged for hand-off with as little physical contact with the family as possible, the A-frame members were redesigned to fold so the unit could be transported flat. This feature also made it possible to collapse the unit and store it out of the way, perhaps under a bed, between HUGS sessions.

The RealSense™ camera was clipped to a bar suspended across two tripods for stability and connected by cable to the USB-3 port of the computer. There was no longer any need to manage the mounting, operation, and data offloading from the ancillary GoPro camera as it was eliminated in the HUGS-2 protocol. To prepare HUGS-2 for collecting data, as in HUGS-1, the parent would open the RealSense™ Viewer software on the study laptop, input resolution settings, and turn on the camera’s RGB and depth sensors. Subsequently, the parent would place the infant in his/her usual, familiar infant seat (versus the hard, standardized seat used in HUGS-1) in front of the RealSense™ and verify that the baby was fully in the camera’s field of view. 

To run the HUGS-2 session, the parent would open one of two custom GUIs (graphical user interfaces) on the laptop. One GUI walked the parent through collecting spontaneous movement video while the other provided guidance on the more complex task of collecting grasp force data (see Figure 10). The grasp force GUI took the parent through the entire data collection process step by step. It instructed the parent when to turn on and turn off the RealSense™ camera. It provided the parent the simple random order for testing light, sound, and haptic feedback modes, which the parent selected by clicking buttons presented on the GUI display. This functionality replaced the analog dial on the HUGS-1 electronics box. The GUI also provided a timing function to signal to the parent when it was time to transition the infant from the present feedback mode to the next. This function replaced the standalone timer that was part of the HUGS-1 process. The GUI also provided a visualization of the magnitude of the forces exerted by the infant’s left and right hands at five second intervals during data collection.

The manual copying of data from the laptop that researchers performed during HUGS-1 was thoroughly automated in HUGS-2. The system was configured to automatically save force data files, as well as the much larger 3D video (.BAG) files generated by the RealSense™ to a password-protected Microsoft OneDrive partition on the study laptop. OneDrive maintained a connection to the Cloud over the family’s home WiFi and uploaded all data created during a HUGS-2 session. Researchers subsequently accessed study data through the OneDrive cloud portal.

#### 4.2.2. Tele-Research Support Plan for Testing under COVID-19 Restrictions

We devised an end-to-end tele-support plan that coordinated contactless drop-off of HUGS-2 hardware, system orientation, coaching, and troubleshooting. We greatly expanded the HUGS-1 user manual into a study website (https://sites.google.com/cua.edu/hugs/home, accessed on 26 February 2023), providing detailed, online, how-to materials, with step-by-step video instructions on HUGS-2 set up and operation. Training was carried out over Zoom with parents simultaneously logged in on the HUGS study laptop and on their smartphones to share views of both the software running on the laptop and the toy bar, electronics box, and other physical system components. 

### 4.3. Back into the Home with HUGS-2

Ethical approval for the HUGS-2 study involving at-risk infants (#20-0060, Home assessment of fine motor development in infants) was granted by the Institutional Review Board (IRB) of the Catholic University of America, Washington, DC, USA, as well as continued from the IRB (IRBear # Pro00011884, Identifying Patient- and Caregiver-Centered Outcomes to Guide the Design and Implementation of Technology for Infant Motor Development Assessment in the Home) of our adjacent clinical partner, Children’s National Hospital. Families with infants 3 to 12 months corrected age, therefore, were identified, informed of the study, and, if they expressed interest, referred to HUGS-2 investigators. Written informed consent was obtained via DocuSign from one of each participating infant’s parents prior to delivery of the HUGS-2 system to the outside of the family home and subsequent parent-managed testing and data collection. The age upper bound specified by the HUGS-1 inclusion criteria was increased from 9 to 12 months of age on the recommendation of children’s site investigators after the beginning of enrollment; therefore, one infant completed her HUGS-2 interactions at 9 months while the remainder completed at 12 months of age.

Entrusting the in-the-moment decision making of HUGS-2 testing to parents surfaced numerous insights that had not been forthcoming when researchers guided HUGS-1 sessions. Though testing under both scenarios took place in the home, implementation differed in several fundamental ways, as described in the sections below.

#### 4.3.1. Increased Focus and Engagement of the Parent

Since the set up and operation were the complete responsibility of the parent, s/he had to develop a greater, more nuanced understanding of the technology and protocol than parents did during HUGS-1, where they typically stepped back and let the research team conduct the session with their infant. Having to make all the decisions in the course of testing also meant that “errors” were made that would not have happened had the researcher been conducting the session and, more importantly, that the researcher would not have actually realized were possible. A prime example of what could be considered a failure of coaching was moving the camera during the session. This action resulted in the need to explore approaches to analysis, as stability of the camera was a pillar of the kinematic algorithm developed to analyze spontaneous movements in HUGS-1 [27]. Other actions that we did not anticipate that added complexity to analysis were bouncing the baby seat during a session and continuing to test on broken toy bars that were suspect, even though they continued to flash colored lights, play music, and vibrate.

Parents proactively alerted the research team to problems with data transmission that we were able to address promptly and stem data loss. These problems were typically downstream of a mismatch between the volume of data being collected and the storage capacity of the HUGS study laptop. COVID-19 forced us to learn the pitfalls of remote transmission well before that functionality problem would have come on the radar under typical researcher-in-home piloting.

#### 4.3.2. More Effective Leverage of Knowledge Only the Parent Has

Leaving it to the parent to figure out the “fit” between the infant and HUGS resulted in more judicious timing of HUGS sessions. Infants were better disposed to interaction with the system as the pressure of a scheduled research team visit was not a factor. This flexibility led to data collection across more varied times of the day than had been the case in HUGS-1. Data were also collected in shorter spurts, over several sessions conducted during the course of a single day, over several days, and occasionally spanning adjacent weeks. 

The parent also adjusted the system setup to the current testing environment. One family moved three times during the nine months their infant was interacting with HUGS-2. They were able to try out different locations for HUGS at their leisure to find the best result. The research team would not have had the time, home knowledge, or access to carry this out well. Parents were also sensitive to when the infant’s seat needed to change to promote optimal HUGS interaction. Incorporating the infant’s current seating strategy over the course of his/her growth and development minimized awkwardness or a misfit between how the growing infant was positioned and the task of playing with HUGS.

#### 4.3.3. HUGS-1 Complexity Resolved Only to Be Replaced by Complexity Emerging from HUGS-2

We attempted to reduce complexity with the development of a GUI incorporating prompts. It needed to be even more user friendly, however. It did not include a prompt to move the camera to the back to capture video of the grasp activity. Parents captured a mix of video from the front and video from the back (ultimately, this “mistake” provided useful information).

As they became accustomed to the process, parents ignored the GUI prompts and forgot to turn off and on the RealSense™ camera between sessions. This problem was retrievable on the backend but resulted in very large video files and subsequent lengthy upload times across the family’s WiFi. The lengthy upload time predisposed families to adopt the “fixed system” versus the “store-away” approach we assumed they would prefer, given that the HUGS-2 A-frame folded flat, unlike the HUGS-1 frame that maintained its wide footprint. Most families found a place to leave HUGS-2 up and ready for testing during the months it was in their home. 

#### 4.3.4. Robustness

Unsupervised in the home, the system was put through its paces and proved less than robust. The system was replaced multiple times for each family, introducing opportunity for calibration drift. Parents proved resourceful at implementing home-made patches. As the study progressed, the research team started to incorporate appropriately sized screwdrivers and Allen wrenches into the HUGS-2 delivery package, along with duct tape. The persistent problem of breakage led to in-progress design changes, principally through experimentation with different materials for 3D printing of A-frame components and permanently affixing the toy bar to the frame. The ability to swap it out via a nine-pin connector was an artifact of the multiple-toy design of HUGS-1. We left this original design feature as we recognized the desirability of novelty and ultimately envisioned having multiple toys as an optimal condition. We did not realize infants would be so adept at disconnecting the toy bar during play!

#### 4.3.5. System Footprint Dilemma

Making the system collapsible allowed for easier packaging, transport, and hand-off. However, parents were reluctant to disassemble the system between sessions. Leaving HUGS-2 constantly at the ready was a factor in breakage, with inadvertent incursions from pets and toddler siblings. One family who did feel the need to collapse HUGS-2 and store due to space constraints withdrew after three sessions due to perceived burden.

### 4.4. Outcomes

Of the six families enrolled during COVID-19, two dropped out, one soon after HUGS-2 package drop-off and before Zoom orientation, the other after completing three sessions. The other four families adhered to the end of their enrollment periods, six for a full nine months. Sessions missed were largely the result of breakage of the bar toy and/or frame (plastic components instantiated via 3D printing) and logistical problems in shipping or delivering replacements. A large gap occurred in one family’s testing due to the family’s coming down with COVID-19. As of September 2022, in total, 25 spontaneous movement videos and 651 recorded grasp events from five infants (three female) at risk for neuromotor delay were available for preliminary analysis. See Appendix A for a detailed treatment of how spontaneous movement and grasp analyses (respectively) were carried out and the results. Due to missing data and unequal timepoints, an LLM was selected for the analysis. Due to the small sample size, α was set at 0.1 for all statistical tests attempted.

#### 4.4.1. Spontaneous Movement Summary

To compare the spontaneous movements of HUGS-2 at-risk infants to those of HUGS-1 TD infants, the average velocity and peak velocity of both right and left elbows and total path lengths of the left and right wrists were calculated for at-risk infants and de novo for TD infants applying a uniform kinematic approach (see Appendix A for the de novo methodology. It applied a different approach than that described in our previously published work [27] to facilitate comparison of HUGS-2 versus HUGS-1 kinematics). Again, an LLM [28] was chosen for conducting the analysis with α set at 0.1 in consideration of the small sample size.

All the following kinematics parameters increased significantly with age for HUGS-2 infants: the average velocity, peak velocity, and the total moving path lengths of both the left and right elbows and the wrists. Comparing the changing trends of these parameters between the HUGS-1 (TD) and HUGS-2 (at-risk) groups, we detected no significant interaction effect of infant age and group. However, which group an infant belonged to did show a significant effect on the average velocity of the right elbow, the moving duration of the right elbow and wrist, and the total path lengths of both elbows and the right wrist. The magnitudes of these parameters were significantly larger in the HUGS-2 versus the HUGS-1 group. Table 3 provides exemplar plots, illustrating the differences in total elbow movement across both groups of infants from three to nine months of age.

#### 4.4.2. Grasp Events Summary Outcomes

As we reported in the pilot of HUGS-1 with TD infants, most grasp-related outcomes (i.e., grasp frequency, accumulated grasping time, peak and average grasp force, the percentage of bi-manual grasps, and the percentage of grasps on the inside of each arm of the bar toy, i.e., closest to the supporting flange) increased with age. Conversely, the force covariance of variation (CV, the standard deviation of the grasp force normalized by the mean force) as well as the percentage of grasps on the middle section of each arm of the bar toy decreased with age. Table 1, pane c illustrates some of the different grasp locations and positions that might be observed during a HUGS session.

The percentage of grasps on the ends of the bar toy and those crossing the midline did not show a significant change with age. The R-ratio for grasp frequency, i.e., the number of grasps with the right hand divided by the total number of grasps, also showed no change over the course of HUGS-2 testing.

We detected an interaction effect (*p* = 0.063, α = 0.1) for the magnitude of mean grasp force between infant age and whether the infant was part of the TD (HUGS-1) or at-risk (HUGS-2) group. This suggests that as age increased, the average (mean) grasp force exerted on the bar toy by the HUGS-2 at-risk babies increased significantly faster than it did in the HUGS-1 TD babies (see Table 4, Mean Grasp Force (b)).

## 5. Discussion

The preceding sections detailed the evolution of HUGS, a home-based system whose purpose is to empower families to monitor their infants’ hand use development where there is concern for possible neuromotor delay. The pilot of the initial prototype, HUGS-1, took place in the home but largely under the direction of the HUGS research team, with parents and other family members in the role of facilitators. Testing was cut short prematurely by the blanket societal lockdown precipitated by the COVID-19 pandemic. When testing resumed on the next-generation, lockdown-friendly HUGS-2 system, the relationship between researchers and family was reversed. The HUGS-2 testing process was under the immediate control of families, with researchers taking up the role of facilitators. 

COVID-19 precautions had the effect of authentically moving the parent into a position of parity with the research team, which was not the case when researchers conducted HUGS sessions in the home. The parent would be helpful but largely observe. Working autonomously in the home under conditions of lockdown, the parent had to assume the position of decision making as to how HUGS-2 testing would proceed on any given day, with the research team consulting virtually and discussing next steps after the fact. 

### 5.1. Knowledge Acceleration

The empowerment of the parent surfaced HUGS use factors that the research team had not considered. Some parental decisions, such as matching HUGS time to the infant’s receptive time, were a great help in the testing process. It was our intention to test only receptive, versus sleepy or cranky, babies, but that did not always happen when researchers made scheduled visit to the home to run HUGS-1 sessions. Other decisions, such as moving the RealSense™ camera during a play session, may have had a detrimental impact on data quality. The research team would not have made this error, our deeper understanding of kinematic algorithm development being comparable to the parent’s deeper understanding of the infant’s shifting circadian cycles. Given the procedural variance observed between researcher-directed HUGS-1 testing and parent-directed HUGS-2 testing, the differences found between TD and at-risk grasp force and movement patterns are suspect though useful for hypothesis generation and informative relative to future home system design.

Extreme partnership with parents brought about by the demands of COVID-19 resulted in deepened appreciation for nuance. Knowledge surfaced sooner rather than later, though perhaps discouraging in the short term, is highly positive for improving the HUGS system and procedures. Our experience is reminiscent of the “fail forward fast” paradigm [31]; one must know about a problem to seek a solution.

### 5.2. Parallels between Pediatric Home Technology Research and Home Clinical Care

There are distinct parallels between our experience of conducting home system research with families pre- and mid-COVID-19 and that of therapists providing services to families in similar situations: those with children at risk for or experiencing neuromotor delay. Pre-COVID-19, it was not uncommon for early interventionists, therapists who provide services for children in their homes, to be left to work with the child while the parent dealt with other tasks. After lockdown, when therapy shifted to telehealth, the parent had to be engaged, an active partner in the home therapy program, as young children do not have the knowledge or skill to manage interactions over telehealth autonomously [17]. This scenario parallels our experience in HUGS-1, where parents often seemed tentative, as if they did not want to “interfere” with the HUGS session. On the contrary, during HUGS-2, parents had to provide the initiative for each session. It seems ironic that in both cases, that of clinicians and that of home researchers, having to interact at a distance through the parent enhanced the relationship and provided deeper insights that supported more appropriately designed interventions in the case of the tele-clinician and more appropriately designed technology in the case of the tele-researcher. 

We observed the principles of the relationship-driven model of family-centered care (RDFCC) play out in our experience working with HUGS-2 families. We propose a parallel paradigm for researchers working with families, particularly in the home. It might be called relationship-driven, family-centered research (FDFCC). It differs from the PSUE framework [21] presented in the Introduction in its focus on the family versus the individual. Family focus is particularly important, as previously noted, when the patient or subject is a child. The principles translate as follows. We offer reflections from the HUGS-2 experience for each principle.

#### 5.2.1. Parity of Parent and Researcher

When the researcher is hands-off and at a distance, empowerment of the parent is a natural consequence. Empowerment is a consequence of parity. We noted increased focus and engagement of the parent in HUGS-2 versus HUGS-1, with more effective application of baby- and family-specific knowledge that only the parent has. When we compare the preliminary grasp force and kinematic data produced by HUGS-1 and HUGS-2, we question why they are so different. Do they point to a real difference between these TD and at-risk infants’ development? Is it possible that the different testing conditions influenced the results? 

Parents’ more insightful choice of times for testing HUGS-2 with their babies may have reflected greater readiness to engage and account for their greater forces, velocities, and arm movement path lengths. Conversely, the more accommodating seating choice provided to HUGS-2 versus HUGS-1 infants may have increased their comfort and held them in a position more conducive to bringing both hands to bear to play with HUGS. Again, the parent bouncing their baby in the seat while the baby interacted with the bar toy also resulted in greater forces. 

#### 5.2.2. Coaching Model 

The precise definition of coaching is controversial. However, a high-consensus characteristic is working collaboratively versus instructing or telling. In the clinical model of coaching, the therapist listens to understand what the family’s goals for therapy are and offers suggestions consonant with family priorities. Conducting HUGS-2 orientation over Zoom provided an ideal opportunity for coaching-in-the-home technology research arena. The parent’s hands, not ours, performed the set up and the parent’s mind was, therefore, active and engaged in thinking about how it would all work together. Researchers could see the parent hesitate or struggle, ask her to share her thoughts, and find a solution together.

#### 5.2.3. Prioritization of Goals of Parent, Baby 

The parent was expert in prioritizing the “goals” of the baby to optimize a happy HUGS-2 session. A key area where the research team failed to understand the family’s priorities was in our assumptions about the relative inconvenience of leaving HUGS-2 set up and ready to go between sessions and breaking it down to store away. That the disassembly was more burdensome for the parents led to one family’s withdrawal from the study and to breakage between sessions because the system was, in fact, in someone’s way.

## 6. Conclusions

We have described how our knowledge about home infant technology and supporting processes accelerated due to the intense engagement of families running the HUGS-2 autonomously in their homes due to COVID-19 precautions. Our next step will be to translate what we have learned into an improved HUGS system and protocol for use. The HUGS research team is currently in a new spiral of development focused on producing HUGS-3, a hand use and grasp sensor system that incorporates lessons learned from both previous efforts, HUGS-1 and HUGS-2. Though the COVID-19 pandemic appears to be on the wane, moving forward, we will apply what we have learned relationally and continue to apply our proposed RDFCT model in practice, carefully considering and applying the best of what virtual and in-person interactions have to offer in working with families. We will also apply what we have learned toward streamlining processes and platforms to create the best fit possible of our technology to families with infants at risk for delay. We distributed requirements for HUGS-3 internally in September of 2022. Appendix A provides a detailed treatment of the current requirements, starting on page 9. Areas to address are recounted briefly below in Section 6.1, Section 6.2, Section 6.3, Section 6.4, Section 6.5, Section 6.6 and Section 6.7. Table 5 summarizes the component changes that will be realized in HUGS-3 versus HUGS-2, an ongoing evolution.

### 6.1. Footprint

Making the A-frame collapsible did not have the desired effect as parents were reluctant to take the system down once it was assembled. The size of the system became a clear issue during HUGS-2, when it was left in the home for the duration of testing (up to nine months). Because it was in the way of family traffic, the rate of breakage and corresponding system downtime was high. Our current thinking is to dispense with the frame all together, put the toy bar in the baby’s hand, and collect grasp signals wirelessly to an app using Bluetooth technology. Figure 11 provides a screenshot of our current Hugs App prototype and the toy bar (without its baby-friendly covering) that it regulates.

### 6.2. Complexity

The introduction of a GUI to coordinate data capture added automation to the process but did not necessarily reduce its complexity. Steps to set up and run a HUGS-2 session were not sufficiently reduced compared to those required for HUGS-1, though automated processes (i.e., clicking on a button) replace manual ones (i.e., turning a dial). Families still made errors, largely through ignoring the prompts. The greatest simplification in HUGS-2 versus HUGS-1 came in the form of automated data upload to the cloud through leverage of a commonly available commercial application (Microsoft OneDrive). The multiple hard (plug-in) connections that had to be made to make HUGS-2 operational were daunting to families, precipitating a desire, as previously noted, to simply leave HUGS set up and (in theory) ready to use across the multiple months it was in the home. A smartphone app, as described in Section 6.1 above, with a freestanding, wireless toy addresses complexity. Our current HUGS-3 prototype uses Firebase to upload to the Cloud versus OneDrive.

Another aspect of HUGS-2 complexity was incorporating both grasp and kinematic assessment in the same platform. The current HUGS-3 prototype separates these features. Its operational simplicity compared to HUGS-2 and HUGS-1 can be seen in a comparison of its operational view (Figure 12) with that of the prior HUGS schemata (Figure 5 and Figure 10). We plan to leverage a commercial off-the-shelf wearable camera, initially with its own smartphone app, to capture movement data.

### 6.3. Durability

Breakage of the frame, and of the bar toy itself, was a frequent problem. Most issues of fragility were a consequence of 3D-printed components, particularly the pivot brackets that held the legs of the A-frame and the T-junction of the bar toy. A wireless toy that resists being pulled apart by the infant, along with integration of a commercial off-the-shelf smartphone, supports a requirement for durability.

### 6.4. File Handling

Transfer of force data from FSR to Arduino to GUI to laptop storage to cloud storage was robust. Each feedback mode data capture took the form of numerical strings saved in .txt format. Hence, file sizes were minimal. This characteristic makes force data a good candidate for Bluetooth transmission from toy to smartphone and for Firebase-managed transmission from the phone to the Cloud. 

In the case of 3D video data, .BAG files averaged five GB each, making transmission from the laptop to the Cloud a lengthy process. We are continuing to explore robust ways of transferring 3D data and falling back to 2D data for limited applications. 

### 6.5. Force Read Accuracy

The FSR used to detect infant grasp force in HUGS-2 is a flat strip sandwiched inside two, 3D-printed half-round pieces to form a cylindrical bar for the infant to manipulate. As previously discussed, it is possible that the differences in forces registered by infants using HUGS-2 versus HUGS-1 are the result of drift or damage, considering that systems we left in the home were continuously subject to breakage. We are currently working on a more robust sensor design for the freely manipulable wireless toy in HUGS-3.

### 6.6. Ability to Measure Criteria Underlying Clinical Assessment of Hand Use

As previously mentioned, in the time since HUGS development and testing have been ongoing, validation data for an assessment of infant hand use, the HAI or Hand Assessment in Infants, were published [32]. The HAI presumes that infants will be able to freely manipulate and maneuver the toys they grasp. The toys that make up the HAI “kit” are untethered, in contrast to the HUGS concept where toys are suspended from a rigid bar. Preliminary analysis of grasp data from HUGS-2 revealed very little vertical movement since infants’ hands were preoccupied with the bar toy, which they explored by moving their hands across its surface horizontally. Figure 13 shows supination and pronation around the toy bar, with a characteristic limited range of vertical movement.

### 6.7. Stimulus

As previously noted, analysis of data from HUGS-1 did not find any difference in grasp force or duration for the different feedback modes: visual, auditory, and tactile. The three modes were retained in HUGS-2 since no difference among stimuli does not mean that the stimulus itself is not useful in attracting a baby’s interest and exploration. Notably, the HAI (Hand Assessment for Infants) manual recommends toys for the assessment toy/tool kit that “make a noise.” Varied textures are also recommended. There is no recommendation for using light as an attractor, however. The HUGS-3 prototype simplifies the design by removing stimulus electronics and using analog ones instead. Various different bar toys incorporating different textures, rattles, and visual attractors are planned.

## Figures and Tables

**Figure 1 healthcare-11-00784-f001:**
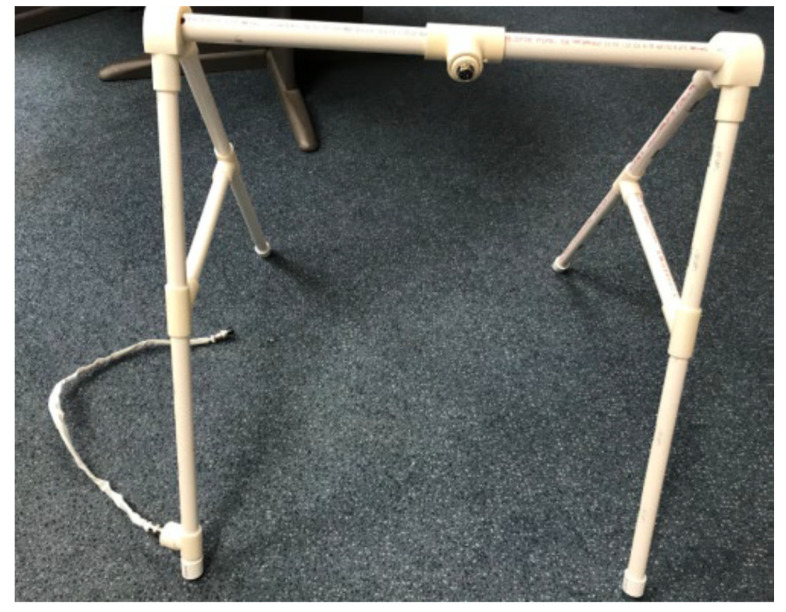
HUGS-1 A-frame for instrumented toys.

**Figure 2 healthcare-11-00784-f002:**
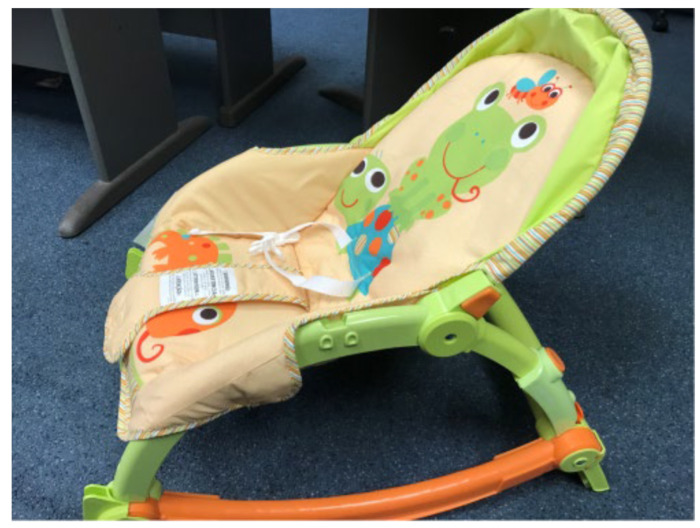
HUGS standardized infant seat.

**Figure 3 healthcare-11-00784-f003:**
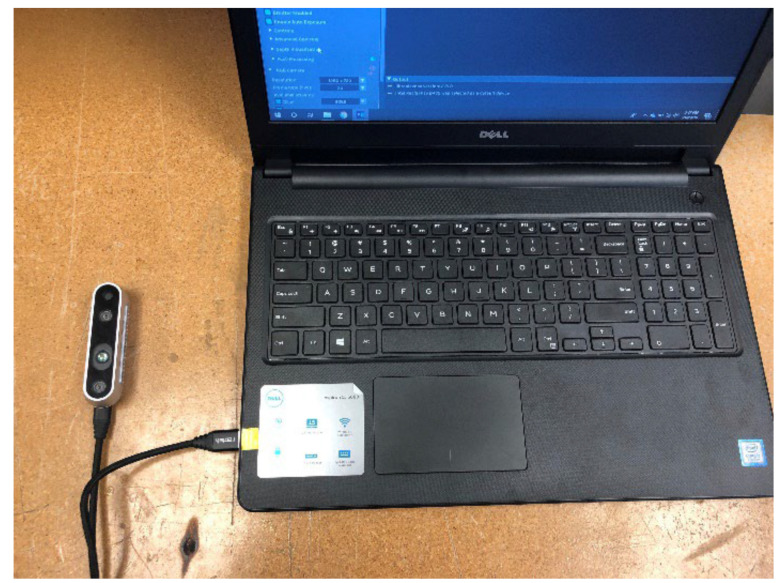
Intel RealSense^TM^ camera and HUGS laptop.

**Figure 4 healthcare-11-00784-f004:**
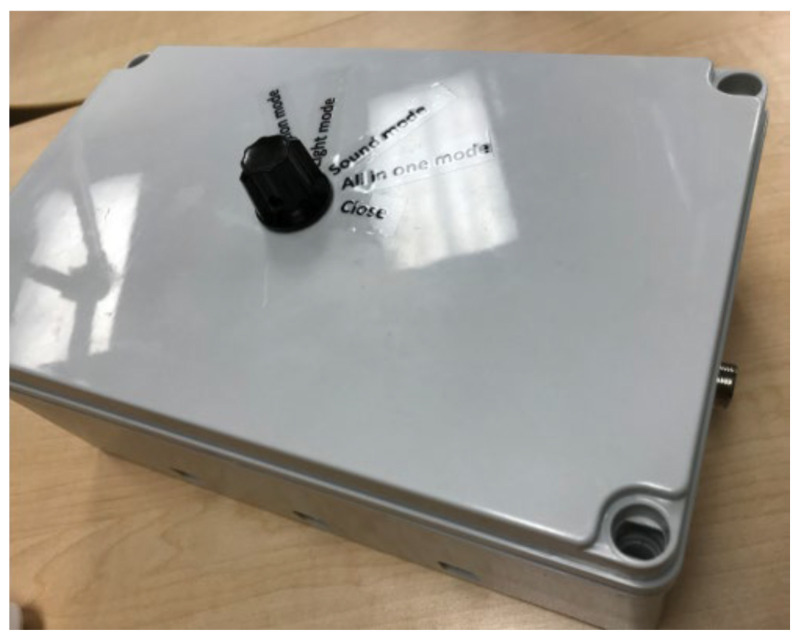
HUGS-1 electronic box with analog dial.

**Figure 5 healthcare-11-00784-f005:**
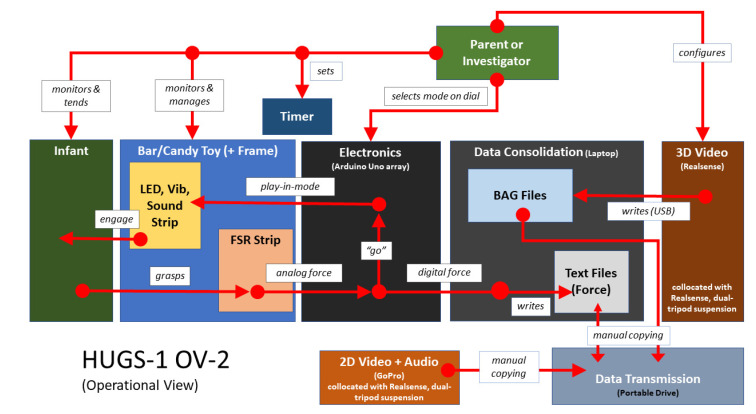
HUGS-1 system components and interoperation.

**Figure 6 healthcare-11-00784-f006:**
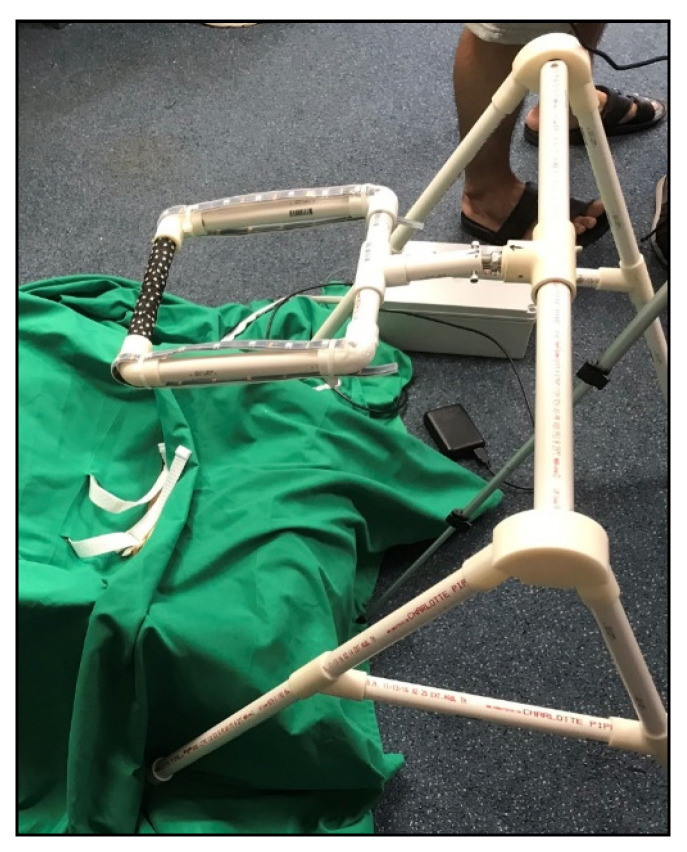
Early rendition of the bar toy with LED light strip on the suspension frame.

**Figure 7 healthcare-11-00784-f007:**
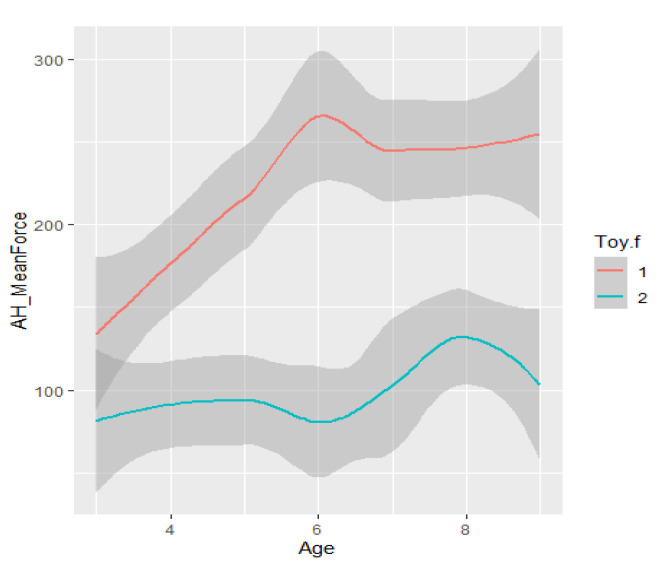
Relative affordance of the bar versus candy toys.

**Figure 8 healthcare-11-00784-f008:**
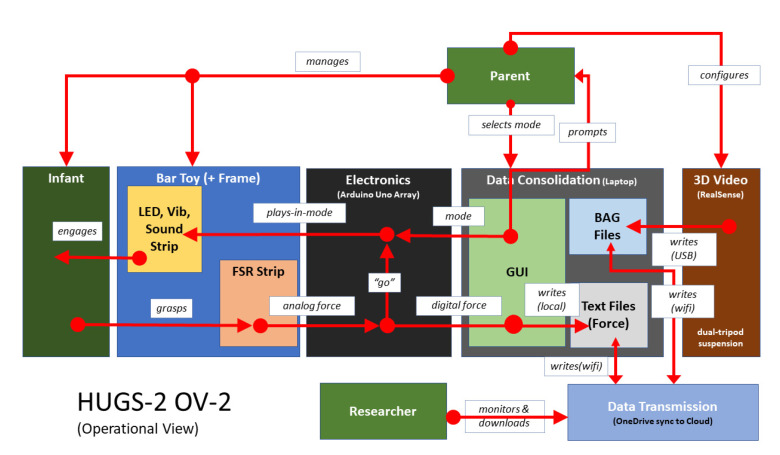
HUGS-2 system components and interoperation.

**Figure 9 healthcare-11-00784-f009:**
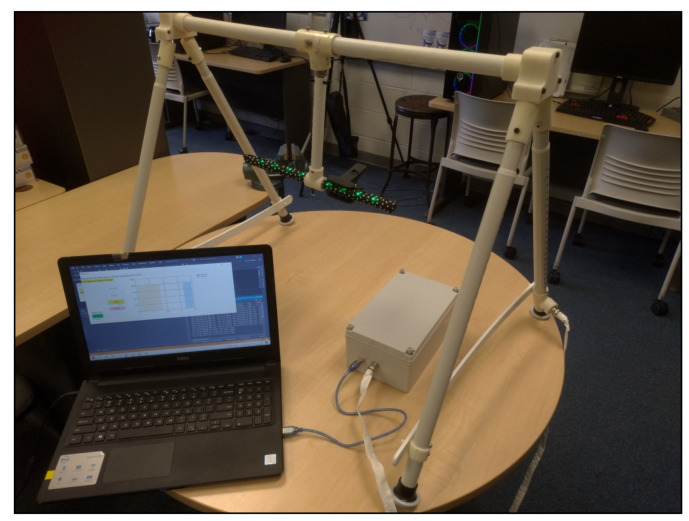
HUGS-2 connections: laptop/GUI, electronics box, bar toy, and frame.

**Figure 10 healthcare-11-00784-f010:**
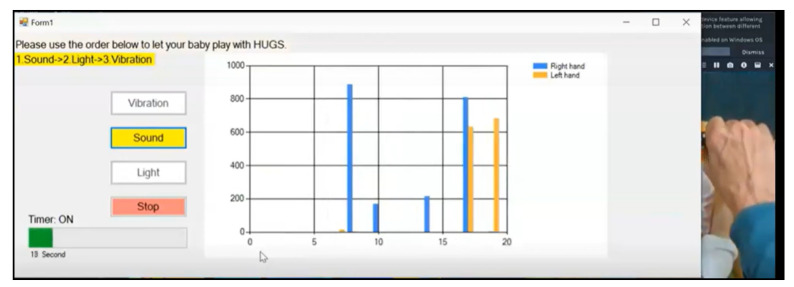
HUGS-2 GUI (graphical user interface).

**Figure 11 healthcare-11-00784-f011:**
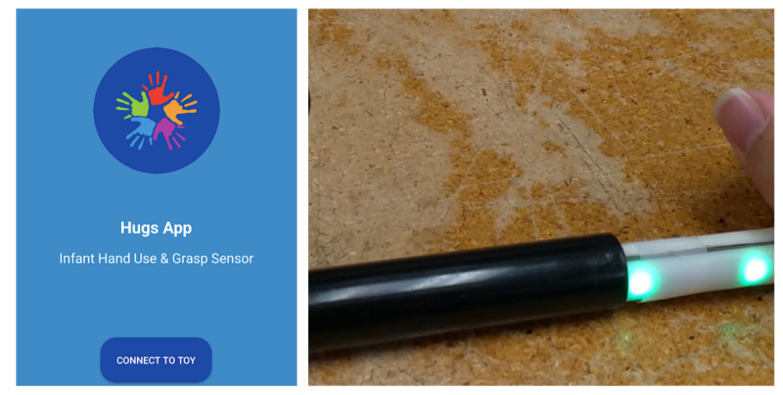
Hugs smartphone app prototype.

**Figure 12 healthcare-11-00784-f012:**
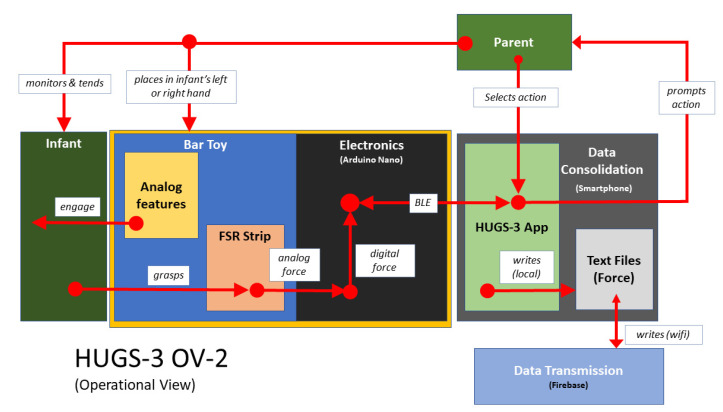
HUGS-3 system components and interoperation.

**Figure 13 healthcare-11-00784-f013:**
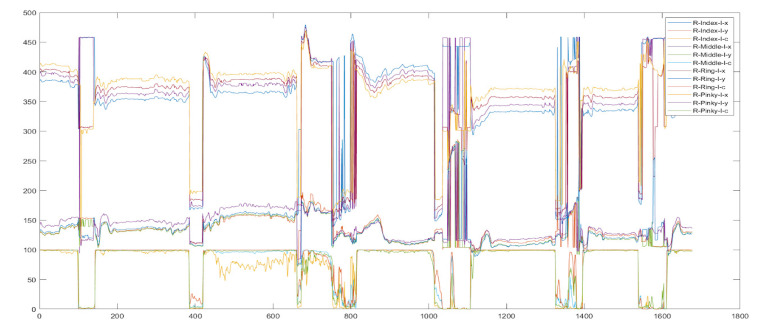
Movements of an infant’s hand on the HUGS-2 toy bar.

**Table 1 healthcare-11-00784-t001:** Infant interaction with HUGS-1.

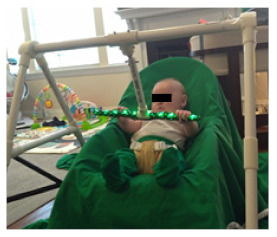	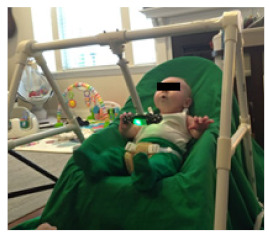
a. Infant playing with bar toy	b. Infant playing with candy toy
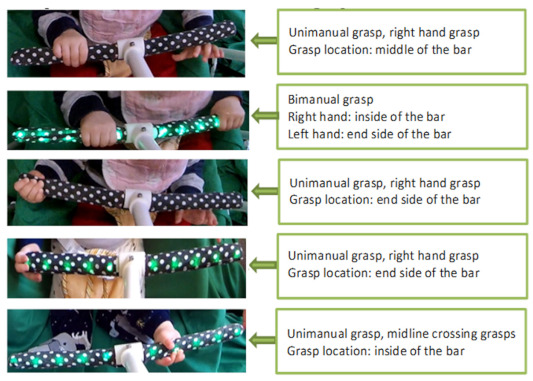	
c. Five common patterns of grasp infants used in engaging with the HUGS-1 bar toy	

**Table 2 healthcare-11-00784-t002:** HUGS evolution: summary design changes HUGS-1 to HUGS-2.

Component	HUGS-1 Characteristics	COVID-19 Motivated?	HUGS-2 Characteristics
Bar/Candy Toy (+Frame)	Rigid A-frame, members fixed in place	yes	Frame collapsible, foldable
	A-frame height fixed to 62 cm	yes	A-frame legs made adjustable (not rolled out to families)
	Toys interchangeable via 9-pin connector	no	Toy permanently fixed to cross bar
	Handling of toy bar restricted to horizontal plane	NA	No change
	Designed to prevent mouthing of toys	NA	No change
Electronics	3-Arduino Uno array for bidirectional signaling, grasp force capture	NA	No change
	Control: analog dial	yes	Control: GUI interface
Data Consolidation	Force data and video data written to files on laptop	NA	No change
Data Transmission	Manually copied to portable storage device	yes	Synced to cloud using Microsoft Windows 10 OneDrive over family WiFi
Video	3D RealSense Camera (no audio)	NA	No change
	Ancillary GoPro Camera (2D with audio)	no	Discontinued
Camera suspension	Horizonal bar supported by two tripods	NA	No change

**Table 3 healthcare-11-00784-t003:** Comparison of HUGS-2 (at-risk) and HUGS-1 (TD) infants’ movement.

Total Path Length, Right Elbow	Total Path Length, Left Elbow
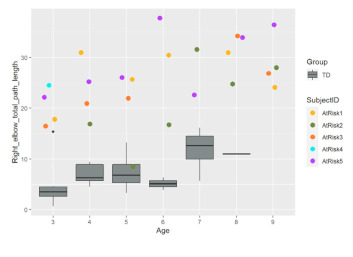	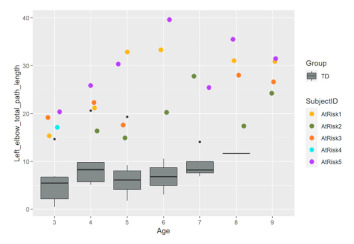
Comparison of HUGS-2 (at-risk) and HUGS-1 (TD, box plots) infants’ movement. Total distance (standardized units, *y*-axis) at each successive HUGS-2 testing session as children developed three to six months of age (*x*-axis).	

**Table 4 healthcare-11-00784-t004:** HUGS-2 (at-risk) versus HUGS-1 (TD) grasp force change over time.

CV, Force Covariance of Variation (a)	Mean Grasp Force (b)
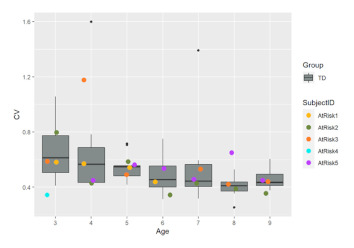	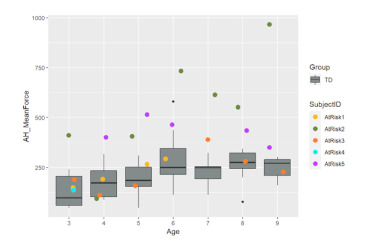
The force covariance of variation (a) decreased as infants grew older for both HUGS-1 (TD) and HUGS-2 (at-risk) study groups. However, the mean grasp force (b) exerted by HUGS-2, at-risk infants per testing session, from three to nine months of age, increased significantly more rapidly (*p* = 0.063, α = 0.1) than did that measured for HUGS-1 (TD) infants across the same time interval.	

**Table 5 healthcare-11-00784-t005:** Continuing evolution: HUG-2 migration to HUGS-3.

Component	HUGS-2 Characteristics	HUGS-3 Characteristics
Bar/Candy Toy (+Frame)	Frame collapsible, foldable	Frameless
	A-frame legs made adjustable (not rolled out to families)	NA
	Toy permanently fixed to cross bar	Toy freed from any tether
	No change	Toy manipulable according to infant’s ROM and hand use ability
	No change	Mouthing afforded and separated from grasp forces
Electronics	No change	Unidirectional transmission only; analog sound, haptic, and visual stimulation
	Control: GUI interface	Smartphone app
Data Consolidation	No change	Grasp forces to phone over BLE, characterized and validated via app
Data Transmission	Synced to cloud using Microsoft OneDrive over family WiFi	Sync to cloud using Google Firebase
Video	No change	Parent vs. video verification (movement and force analysis split)
	Discontinued	NA
Camera suspension	No change	NA

## Data Availability

The data presented in this study are available on request from the corresponding author. The data are not publicly available due to the need to protect the privacy of participating families who might be identifiable from the video that constituted the majority of the dataset.

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
