# Peer review of "Evolution of a System to Monitor Infant Neuromotor Development in the Home: Lessons from COVID-19"

_healthcare, 2023, doi:10.3390/healthcare11060784_

Round 1

Reviewer 1 Report

The article's authors present a system to monitor infant neuromotor development using force sensor-embedded toys and 3D video captured during the interaction of babies with toys. The system is well presented, along with comparing two different versions of the system. However, there is a lack of scientific evidence - e.g., a thorough analysis of the use of both system designs by infants would improve the article's value. 

Rewrite the paragraph of line 76 (pay attention to line 79).

Reviewer 2 Report

Very interesting subject ; suitable for publication.

Please, find my comments below :

I recommend that authors should more summurise the step of the study:

Step 1: Description of the equipment;

Step 2: Description of the start of the HUGS;

Step 3: Description of the difficulties in technical and practical levels;

Step 4: Description of the evolution of the HUGS

In the results part: Keep that the description of detected grip anomalies thanks to this system

Reviewer 3 Report

This research details the evolution of HUGS, a home-based system whose purpose is to empower families to monitor their infants' hand use development where there is concern for possible neuromotor problems. The first stage took place in the home but mainly under the direction of the research team, while parents were in an auxiliary role. The experiment ended prematurely by SARS-CoV-2 pandemic. In the second stage, the process was under control of families, with researchers taking up the role of facilitators.

The written research is long and needs extensive editing of English language. References don't follow the numerical and usual standards. This point is the least notable. Table 2 deserves further editing, as well as tables 3 and 4. A stark cutting in the images should be solved. The same can be applied to language. For instance (sic): 
. what we had learning
. (te Velde et al., 2019)
. was The need to move most service provision to telehealth was a
. to a much simpler system for in the home
. to be implement
. Five, representative infant grasps
. a high-level, view
. selected. (See Figure 6)
. it was in their home. Robustness
